# Advanced Oxidation Processes and Biotechnological Alternatives for the Treatment of Tannery Wastewater

**DOI:** 10.3390/molecules26113222

**Published:** 2021-05-27

**Authors:** Néstor Andrés Urbina-Suarez, Fiderman Machuca-Martínez, Andrés F. Barajas-Solano

**Affiliations:** 1School of Natural Resources and Environment, Universidad del Valle, Cali 760015, Colombia; nestorandresus@ufps.edu.co; 2Department of Environmental Sciences, Universidad Francisco de Paula Santander, Av. Gran Colombia No. 12E-96, Cucuta 540003, Colombia; andresfernandobs@ufps.edu.co; 3Centro de Excelencia en Nuevos Materiales–CENM, Escuela de Ingeniería Química, Universidad del Valle, Cali 760015, Colombia

**Keywords:** wastewater, AOPs, tannery, microalgae, biological process, leather industry

## Abstract

The tannery industry is one of the economic sectors that contributes to the development of different countries. Globally, Europe and Asia are the main producers of this industry, although Latin America and Africa have been growing considerably in recent years. With this growth, the negative environmental impacts towards different ecosystem resources as a result of the discharges of recalcitrated pollutants, have led to different investigations to generate alternative solutions. Worldwide, different technologies have been studied to address this problem, biological and physicochemical processes have been widely studied, presenting drawbacks with some recalcitrant compounds. This review provides a context on the different existing technologies for the treatment of tannery wastewater, analyzing the physicochemical composition of this liquid waste, the impact it generates on human health and ecosystems and the advances in the different existing technologies, focusing on advanced oxidation processes and the use of microalgae. The coupling of advanced oxidation processes with biological processes, mainly microalgae, is seen as a viable biotechnological strategy, not only for the removal of pollutants, but also to obtain value-added products with potential use in the biorefining of the biomass.

## 1. Introduction

Currently, there are various industrial processes that contribute to the deterioration of water quality; most require the use of highly polluting substances for terrestrial and aquatic ecosystems; therefore, it is imperative to apply a treatment before its discharge in different water bodies. The tanning sector allows the transformation of degradable animal leather into non-degradable leather by eliminating impurities and unnecessary materials [1]; however, this process generates different residues that represent a significant environmental problem, requiring specific treatments, which are expensive and inaccessible in many developing countries. [2]. In the global context, Asia together with Latin America and Europe are the main leather producers in the world [3,4] (Figure 1). It is estimated that the budget of this business in Europe is approximately 8 billion euros per year, due to the more than 3000 companies that employ 50,000 people, demonstrating their competitiveness in the global market [5]. In recent years, Latin America and Africa have increased the annual growth rate given the consumption of leather in these places [3]. Italy is the main leather producer on the European continent, owns 60% of the companies and exports more than 70% of the total production in Europe; it owns 15% of the world leather production, and in the European Union, it is the main producer with 65% of the total manufacture. [6]. In Latin America, Brazil and Argentina stand out in this sector; they have a significant number of emerging companies with great competitiveness in the market, exporting their products to countries such as China, Hong Kong, Vietnam and the United States. Globally, Colombia represents 5% of the tanning industry production, and in the Latin American context, it ranks 6th as a producer in this industry [5].

Leather production consumes on average 10–25 m^3^ of water in its different stages and can generate on average 8 to 20 m^3^ of wastewater; this flow may vary depending on the technological development of each industry. According to Nagi et al. [2], this resource is used to transport chemical products from diffusion and in the phase of washing and extraction of undesirable compounds from the leather [7]. In this process, large amounts of wastewater are generated, significantly altering the quality of the water since they not only contain biodegradable compounds such as fats, proteins and carbohydrates, but also polluting compounds such as solvents, additives and toxic heavy metals typical of the process [8]. At least 90% of industries are reported to use basic chromium sulfate as a tanning agent [9,10], chromium (Cr) being one of the most toxic metals used in tanning and, without prior treatment, is released in effluents in toxic concentrations. The regulations of the World Health Organization (2020) recommend a reference value of 0.05 mg/L of total chromium in drinking water; above this concentration, it can exert a genotoxic and carcinogenic effect. High concentrations of chromium (VI) cause the deterioration of ecosystems and have an impact on human health as they cause kidney damage, liver damage, chronic bronchitis, nasal irritation, cancer, and DNA damage among other things [11,12]; these effects have led governments to apply stricter regulations and promote effective treatments to reduce the risk of contamination. Over the years, various techniques for treating tannery wastewater have been studied, focused on the removal of pollutants, particularly chromium. Some of these technologies are chemical coagulation, photodegradation, biodegradation, adsorption, ozonation, electrocoagulation and reverse osmosis [13]; however, the difficulty of implementing them is related to high energy consumption, use of large areas of land, high operation and maintenance costs [14]; in the same way, in some processes, other polluting byproducts are generated [15]. The foregoing has launched various biotechnological processes, including the use of microalgae, bacteria and fungi, as sustainable and economically affordable alternatives [16].

In recent years, microalgae and cyanobacteria have been proposed as a sustainable solution for the removal of nutrients and hazardous materials from wastewater [17,18,19,20]. In relation to the treatment of wastewater from tanneries, studies with various strains are still limited, the most common being *Scenedesmus* sp. and *Chlorella* sp., obtaining positive and promising results [2]. Microalgae have exhibited high tolerance to adapt in this environment, as well as an important role in reducing contaminants. The biomass of the microalgae has shown an absorbent capacity during the chromium elimination process, obtaining important results with *Chlorella vulgaris*, where a reduction of Cr(VI) to Cr(III) was also observed, going from a highly toxic compound to one less toxic, also attributing it to biological (enzymatic pathway) and non-biological mechanisms (glutathione releasing) [21]. Likewise, the use of residual biomass from the process is a way to maximize production of energy (biofuels) and generate byproducts of commercial interest (pigments, lipids, etc.), reducing the costs of the process [22,23,24,25,26,27,28,29]. Various studies have been carried out to identify the advantages of using microalgae in tannery waters. In India, one of the three largest leather producers after China and Italy, high concentrations of Pb, Cr, Zn and Cu were found, and an efficiency of 60–98% of removal of these metals under different conditions was demonstrated, specifically with microalgae [30]. In Brazil, progress has been made regarding the growth of microalgae consortia under different concentrations of tannery wastewater [16].

In Colombia, several research studies have been carried out to mitigate the impacts; among them are the identification and evaluation of pollutants [31]; the use of the *Eichhornia crassipes* plant has been reported in pilot-scale treatment of tannery water due to its capacity to accumulate heavy metals and organic matter [32]; the genotoxic effect of tannery effluents on *Salmonella typhimurium* strains has also been studied, identifying a mutagenic increase and the capacity to generate DNA damage in human lymphocytes [31,32,33]. However, the application of microalgae in tannery wastewater has not yet been developed in depth in this country.

This review exposes the current situation of tannery wastewater treatments, focusing on the implementation of new biotechnological tools and, in more detail, on the use of microalgae as a treatment to reduce pollutant loads and the use of metabolites of microalgae in this process as a source of biofuels or byproducts of industrial interest.

## 2. Pollutant Loads from Tannery Wastewaters

Tannery wastewater is the product of a transformation process from organic matter to non-degradable matter, which requires the addition of compounds and additives that allow such transformation, generating in turn waste highly polluting not only for human health, but also for the environment [34]. These wastewaters have as main characteristics a dark brown color, a characteristic odor due to the presence of volatile organic compounds, organic and inorganic carbon, phosphorus (P), nitrogen compounds (N) [9,35,36,37,38,39,40,41,42], fats and other highly polluting compounds at certain concentrations, such as chemical oxygen demand (COD), biological oxygen demand (BOD), total dissolved solids (TDS), chlorides, sulfates and heavy metals such as Zinc (Zn) and chromium (Cr), among other things [37,43,44,45,46]. Goswami and Mazumder [47] reported a typical characterization of tannery wastewater, where COD concentrations were observed between 500 and 11,500 mg*L^−1^, total Kjeldahl nitrogen (TKN)—200–550 mg*L^−1^, observing that the highest fraction is available as ammonia nitrogen (NH_3_–N), total chromium concentrations (Cr(VI) and Cr(III)) in a range of 5–140 mg*L^−1^, slow biodegradability due to the content of biodegradable compounds is less than 50% and very high total dissolved solids (TDS) compared to total suspended solids (TSS). Different studies have reported high values of BOD, COD and even the presence of Cr in effluents [13,36,39,48,49,50,51,52,53,54,55]. Other compounds have been reported in these wastewaters, finding results of acidic pH between 3.4 ± 0.0351 and 5.96 ± 0.0351 [56,57,58] and basic pH between 8.0 ± 0.4 and 11.64 ± 0.53 [37,41,46,51,52,54,55,59,60,61]; in relation to TDS, typical values can have concentrations ranging from 2355 ± 85 mg*L^−1^ to 10,000 ± 800 mg*L^−1^ [9,37,39,42,61,62]; high concentrations have been recorded that can be between 10,560 ± 78 mg*L^−1^ and 72,400 ± 0.10 mg*L^−1^ [35,40,53,54,55,63]; as for BOD, the average values can be in low ranges from 160 ± 15.8 mg*L^−1^ to 1250 ± 38 mg*L^−1^ [37,39,61,63] and in high ranges that fluctuate between 1500 ± 41 mg*L^−1^ and 6000 ± 30 mg*L^−1^ [9,49,50,51,53,54,55]. In relation to the total chromium concentration, values ranging from 0.83 ± 0.028 mg*L^−1^ to 134 ± 5.8 mg*L^−1^ have been reported [9,35,38,40,61,64], as well as high ranges from 147.4 ± 1.5 to 3800 ± 115 mg*L^−1^ [35,37,44,53,56,57,60], chloride—between 1101.9 ± 1.6825 mg*L^−1^ [65,66] and 1696.6 ± 1.8965 mg*L^−1^ and sodium—with a maximum range of 690.1 ± 1.0504 mg*L^−1^ [67,68,69]. Likewise, differences in the results of raw and pretreated tannery wastewater by conventional coagulation process have been reported, where total solids (TS) results of 10,265 ± 1460 mg*L^−1^ and 6810 ± 110 mg*L^−1^ were obtained. COD was 4800 ± 350 mg* L^−1^ and 1910 ± 174 mg*L^−1^, TKN—225 ± 18 mg*L^−1^ and 203 ± 23 mg*L^−1^, NH_3_–N—128 ± 20 mg*L^−1^ and 120 ± 15 mg*L^−1^, and total chromium—95 ± 55 mg*L^−1^ and 0.55 ± 0.11 mg*L^−1^, respectively, showing that the coagulation process as pretreatment decreases the concentration of some parameters [70,71,72,73]. Table 1 shows different values of physicochemical characterization of tannery wastewater.

## 3. Technologies for the Treatment of Tannery Wastewater

The main technologies for the treatment of tannery wastewater focus only on certain parameters [75], as there is great difficulty in finding a treatment that completely reduces the pollutant load. Figure 2 presents a graphical description of the most common technologies. Since chromium is the main problem in this process, most treatments for these waters focus on the reduction and reuse of this element, followed by treatment for COD, BOD and TDS. These technologies include chemical coagulation processes [76,77] electrocoagulation [78], absorption, advanced oxidation processes [79] and biological processes such as phytoremediation [13,14].

### 3.1. Coagulation

Coagulation has been widely used due to its ease of operation; however, the generation of secondary waste from this process requires the greatest attention [6,78]. This method adds compounds such as aluminum sulfate and ferric chloride, which affect the removal of suspended solids, COD and chromium up to 46%, 37% and 99% at optimal concentrations of the coagulant and optimal pH ranges (7.5), respectively [78]. In addition, it works as a pretreatment method for tannery wastewater as it allows removing chromium and limiting its inhibitory effect on biological processes [79].

### 3.2. Electrocoagulation

Electrocoagulation is an electrochemical process that has the same principle as coagulation but reduces the formation of sludge typical of this process due to the fact that the coagulant is generated in situ by the oxidation reaction of an anode [80,81,82]. The use of mild steel electrodes as anodes under specific operating conditions has generated efficiencies of 82%, 90% and 96% for COD, sulfates and greases [83]. Some processes have implemented hybrid electrocoagulation and electrodialysis systems, showing greater efficiency to improve the quality of treated wastewater, obtaining removal percentages for COD, NH_3_–N, chromium and color of 92%, 100%, 100% and 100% in processes with aluminum electrodes and 87%, 100%, 100% and 100% in processes with iron electrodes [84]. All these technologies improve the quality of the wastewater and allow the removal of polluting compounds; however, there are drawbacks with these treatment systems due to the high production of toxic sludge, the high operating cost and the complicated management of some of them, which are limited technologies in the developing countries [65]. This has led to the incursion of new technologies that are accessible and capable of mitigating environmental impacts, with advanced oxidation processes and biological processes being the main focus. This has allowed the development of research that shows excellent results when combining both processes, such as the synergy between biological treatments combined with electrooxidation [85] and, in other cases, with Fenton reagents [86].

In relation to the removal of Cr(VI), electrocoagulation has been implemented in recent years; in this process, flocculation occurs “in situ” due to the electrooxidation of a sacrificial anode (usually Fe or Al) [87]. One of the advantages of this process compared to chemical coagulation lies in the sludge production; sludge reduction in electrocoagulation is 50% compared to chemical coagulation, showing more environmentally friendly properties. The process consists of transforming (directly or indirectly) Cr(VI) into Cr(III) and then precipitating and separating Cr(III) as a hydroxide [57]. Removal efficiency of Cr(VI) has been reported up to 99% in a pH range of 5 and 8; above this range, the removal efficiency decreases up to 27% while at the pH lower than 5, 50% of Cr remains dissolved and the rest is electrodeposited in the sludge generated; therefore, the control of the pH of the solution is a variable that affects the process. It is known that chromium deposition is possible from solutions based on much less harmful Cr(III) compounds which are obtained by electrochemical processes. These electrolytes can be a real alternative; however, the electrochemical reactions that take place in the electrodeposition of chromium from Cr(III) salt solutions are complicated and their understanding still needs to be studied. In an electrodeposition process using electroplating, the removal of chromium ions from tannery wastewater was evaluated in a synthetic trivalent chromium solution; 96.5% of the total chromium content was removed in the untreated effluent [88]. Finally the route dictated by the thermodynamics of the multistep reduction of Cr(III) to Cr evidences that metallic chromium is probably deposited through the discharge of electroactive hydroxo complexes of bivalent chromium that form in the near-cathode layer due to the dissociation of water molecules [89,90], but it is still necessary to evaluate this process more extensively in tannery wastewater.

### 3.3. Advanced Oxidation Processes (AOPs)

Advanced oxidation processes involve the production and application of highly oxidative radicals, primarily the free hydroxyl radical (OH), capable of selectively degrading recalcitrant contaminants to a less harmful state. Figure 3 represents the most common advanced oxidation processes such as Fenton and photo-Fenton processes, ozone-based processes, photocatalysis, oxidation processes and hydrogen peroxide-UV processes, among other things [8,50,91,92,93,94].

Due to the ability to oxidize a wide range of micronutrients, these processes have been very useful in wastewater treatment; among the main ones is Fenton [95,96,97,98] which uses ferrous iron (Fe^2+^) to decompose hydrogen peroxide and form an OH radical before reducing it again [99]; however, one of its main limitations is the disposition of large amounts of ferric ions in the mud, which can be overcome by using the reagent photo-Fenton which is a cyclical process and regenerates the Fe^2+^ ion [91,100]. The Fenton process has obtained important results in the treatment of tannery wastewater, 93% COD, 98% BOD and 62% chromium were removed when combined with a biological process using *Thiobacillus ferrooxidans* [86]. Another advanced oxidation method is ozone, which has been implemented in wastewater treatment due to its ability to reduce color, synthetic aromatic compounds and persistent organic pollutants (POPs) [101,102,103,104,105]. This process has not only demonstrated its effectiveness in pollutant removal, but also works well with biological degradation processes as the integrated treatment significantly improves performance [35,106].

Electrochemical processes have had great relevance in recent years [83,107]; this technology generates oxidizing agents, destroying organic compounds until their mineralization, using methods of anodic oxidation, photoelectrocatalysis [108,109] and electro-Fenton [110,111,112]. This method allows the elimination of toxic compounds, nutrients and non-biodegradable organic compounds from tanneries with high levels of salinity and organic pollutants due to the implementation of different electrodes under the activity of direct and indirect oxidants [113]. Table 2 shows different AOPs used in the treatment of tannery wastewater.

In AOPs, the reaction of OH and the various pollutants present in tannery effluents results in mineral end products that produce inorganic ions and carbon dioxide [114,115]. The efficiency of each process depends on the physicochemical characteristics of the pollutants present in tannery effluents, as well as on the generation of hydroxyl radicals. The generation of these radicals can be achieved by different processes. Ozone is a technology that has been reported in the treatment of dyeing water, reaching removals of 90–98% COD and 96% color [104,105]; the efficiency of the process depends largely on pH; at acidic values < 4.5, the reaction is direct, molecular ozone dominates the reaction being selective mainly in the destruction of chromophore groups, while at the pH > 7, ozone decomposes, generating OH which is less selective and has a higher oxidation potential; the ozone flow rate is another important variable, the percentage of removal is directly proportional to the ozone flow rate with respect to time; increasing the ozone rate increases the removal efficiency. It has been reported that excess ozone can lead to the formation of residual H_2_O_2_, allowing the wastewater concentration to be increased so that the COD present can be degraded by the excess H_2_O_2_ [114].

In relation to photocatalytic processes, photo-Fenton is one of the most studied in tannery effluents, and has achieved removals of 70–90 % COD, 86–98% color [100,111] and 90% Cr [114]; the efficiency of the process depends largely on the pH of the solution, the optimal range of higher catalytic activity is 2.8–3.0, pH values > 5 generate ferric hydroxides that reduce the reactivity of OH, while at values below 2, complex iron species are formed that react more slowly with H_2_O_2_, decreasing the efficiency of the process [116]; the amount of ferrous ions generated also affects the process; excess concentrations in the solution can generate precipitates, increasing the TDS concentration [117]. Finally, the irradiation time is another variable that affects the process; this should be as low as possible to minimize energy consumption without affecting process efficiency. The UV/H_2_O_2_ system has great relevance in the treatment of tannery effluents, the main reason being the absence of sludge production and significant COD removal in very short reaction times [118].

The effectiveness of the UV/H_2_O_2_ process for the degradation of complex compounds present in these effluents depends on several factors. The pH affects the reactivity of H_2_O_2_ as well as the generation of the OH radical, therefore, a pH of 3–5 is recommended to implement the UV/H_2_O_2_ process. The type of the UV lamp is another important variable in this process; the selection of the waves generated by the lamp is a design parameter that defines the efficiency of the system. The medium-pressure ultraviolet lamp (MP-UV) and the low-pressure ultraviolet lamp (LP-UV) are the two types of lamps used in the UV/H_2_O_2_ system. The MP-UV lamp is usually the most widely used as it is capable of emitting a broad spectrum of waves much faster than the LP-UV lamp, allowing a rapid dissociation of peroxide radicals resulting in a direct photolysis that allows a faster degradation of the pollutants present in tannery effluents [110]. Temperature is an important factor in the UV/H_2_O_2_ system; at room temperature, the reaction of the peroxide with the pollutants present in the tannery effluents is lower, hence, it is required to accelerate the process by increasing the temperature (40 °C and 60 °C), allowing the generation of OH from H_2_O_2_ and increasing the reactivity of these radicals towards pollutants [119].

Cavitation is a phenomenon that results in the generation of highly reactive free radicals, releasing large amounts of energy and creating intense turbulence in the liquid. Depending on the cavitation mode, four different forms are distinguished: acoustic cavitation, hydrodynamic cavitation, cavitation, optical cavitation and particle cavitation. It has been reported that acoustic cavitation and hydrodynamic cavitation have proven to be effective techniques for the treatment of tannery effluents [114,120]. The process parameters that affect acoustic cavitation are as follows: ultrasonic power; since the amount of OH generation and the energy dissipated depend on the ultrasonic power and the immersion depth of the probe, it is essential that the immersion depth of the probe is adequate for the propagation and dissipation of the wave. As for hydrodynamic cavitation, factors such as pH, temperature and inlet pressure affect the removal efficiency of parameters such as COD and color [121].

#### Energy and Cost Considerations in AOPs

The selection of an AOP for wastewater treatment is not only based on the removal efficiency, although it is important, the energy consumption or efficiency of an AOP has taken great relevance when selecting a process. The electrical energy per order (EEO) is an indicator used to determine energy efficiency [122], defined as the amount of kWh required to reduce the concentration of a pollutant by an order of magnitude in a given volume of treated wastewater (kWh m^−3^ log^−1^) [123]. This parameter has been used to analyze and compare different AOPs in the treatment of pollutants present in wastewater; the reported EEO values are in the range of 0.1–15 kWh m^−3^ log^−1^ for processes such as ozonation, UV/O_3_, UV/H_2_O_2_, EB, photo-Fenton, O_3_/H_2_O_2_; some processes such as UV/catalyst, microwave and ultrasound report high values in the range of 150–8000 kWh m^−3^ log^−1^ [124]. The EEO value varies in relation to the process efficiency: the higher the process efficiency, the lower the calculated EEO value.

Another aspect that affects the EEO value is related to the characteristics of the wastewater and the types of pollutants present; a very complex wastewater matrix with several pollutants will obtain higher EEO values compared to simple matrices or wastewaters with only one pollutant [121]. Wardenier et al. [125] conducted an investigation comparing UV, O_3_ and H_2_O_2_ efficiency for the treatment of micropollutants such as atrazine, alachlor and bisphenol and found that energy consumption and treatment cost were in the following order: UV > UV/H_2_O_2_ > UV/O_3_ > UV/O_3_/H_2_O > O_3_/H_2_O_2_ > O_3_, while the order in relation to removal efficiency was O_3_/H_2_O_2_ > UV/O_3_/H_2_O_2_ > UV/O_3_ > UV/H_2_O_2_ > UV/H_2_O_2_ > O_3_ > UV. In an azo dye treatment process, when implementing ultrasound with AOPs to achieve 90% removal, the order in relation to cost and energy consumption was as follows: US > photocatalysis > UV > UV/H_2_O_2_ > UV/O_3_ > photo-Fenton > O_3_; in relation to the percentage of removal, the order was as follows: photocatalysis > photo-Fenton > UV/H_2_O_2_ > O_3_ [125]. Performing these calculations and comparing by technology provides information for process scale-up as well as for sustainability analysis [126,127,128,129,130]. Table 3 shows a comparison in terms of energy consumption and cost of different AOPs.

### 3.4. Biotechnological Conversion of Tannery Wastewater

Currently, the capacity of biological processes in wastewater treatment has been demonstrated to be more environmentally friendly for the elimination of organic compounds and micropollutants compared to other technologies [131]. However, the high concentration of pollutants makes these treatments difficult to implement since the levels of toxicity and the impacts generated on the cells make their control and the adaptation of the species to this process complicated. These difficulties have been overcome in recent years, where different microorganisms have been proposed as an efficient alternative and an opportunity to establish bioremediation centers and economic biorefineries to recover resources and promote sustainable economic development for the treatment of wastewater and the removal of heavy metals from water [131,132]. Studies on these processes use biological agents such as bacteria, microalgae and some fungi, among others [7,133].

#### 3.4.1. Bacteria

Within the biotechnological processes implemented for the treatment of tannery wastewater, the use of chromium-resistant bacteria has been reported as an economical and ecological alternative for the detoxification and bioremediation of these effluents due to the ability to adapt to exposure stress, developing various mechanisms of resistance and adaptation [132]. One of the processes that have been implemented is the batch reactor systems (SBR) combined with respirometry; the decrease in COD concentration by 74–88% has been reported in 12- and 24-h operating cycles, which is more efficient than when continuous aerobic systems are used; however, to improve the removal of the remaining COD fraction, the development of more ecological chemical products for tanning is suggested [134]. On the other hand, in activated sludge systems [135], positive results are shown in the removal of BOD_5_ and COD, of 90% and 80%, respectively, with specific operation requirements in suspended solids of mixed liquor (MLVSS) of 3500 mg/L maintaining aeration time of 12 h [136]. In treatment systems with plants, pilot-scale experiments of tannery waters from south of Bogotá (Colombia) have been developed through the use of *Eichhornia crassipes*; the use of wastewater occurred in proportions of 40% wastewater/60% distilled water, and 60% tannery water/40% distilled water, with initial chromium values of 7480 mg/L and 12,200 mg/L, respectively, obtaining chromium removal of 61% and 51%, although the results are not significant. The discharge standard was met due to the high initial amount of chromium [32].

The use of different species of bacteria has been reported with the ability to degrade tannery wastewater. It has been found that different species of *Bacillus* such as *Bacillus* sp. [137], *Bacillus flexus* [138], *Bacillus aquimaris* [54], *Bacillus cereus* [139] and *Bacillus subtilis* [140] can remove up to 85% COD and 54–95% Cr present in these effluents. In relation to species of the genus *Pseudomonas*, it has been reported that *P. aeruginosa* [54,139] and *P. putida* [141] can remove up to 98% COD and 93% Cr. Species of the genus *Halomonas*, *H. maura*, *H. pacifica* [142] and *Halomonas* sp. [143], have been found to remove COD in the range of 76–96%. *Desulfovibrio desulfuricans* species [144], *Desulfovibrio* sp. [145], with the ability to remove up to 65% Cr and 85% COD has been reported. *Microbacterium arborescens* species [54] *M. testaceum* [146] can remove up to 68.4% COD and 99% Cr. Different enterobacteria have been reported for the biotreatment of tannery wastewater, *Enterobacter* sp. [54], *Serratia marcescens* [147] and *E. coli* [148]; they have the ability to remove up to 63% COD and up to 54% Cr. The groups of nitrifying bacteria [149] and denitrifying bacteria [150] have also been reported in the treatment of these effluents, reaching removal percentages of 80% COD and 64.4% Cr and 98.3% COD and 88.5% Cr, respectively. Other microorganisms have been found with the ability to treat tannery effluents: *Shewanella xiamenensis* [151] has exhibited an ability to remove 80% COD and 73% Cr; *Acidithiobacillus thiooxidans* [152] achieved removal of 95% Cr, while species of *Cellulosimicrobium* [37] have achieved removal of 99.3%. All this has led to the bioprospecting of new species that allow better results.

#### 3.4.2. Utilizing Tannery Wastewater in Microbial Fuel Cells

Chromium exists in various oxidation states, and, among them, Cr(VI) is considered pollution with a high degree of toxicity [153]. The process of removing Cr in wastewater such as tannery wastewater is energy-intensive [154]; therefore, current studies have focused on seeking to add value to this process to mitigate the high cost of this treatment [155,156,157]. One of the technologies that has gained great relevance is the microbial fuel cell (MFC); this process is an interstate alternative for the treatment of Cr(VI), in which Cr(VI) is reduced in the MFC through bacterial activity [158], which results in the generation of electricity, offsetting the cost of the treatment process [159,160,161]. MFCs are devices that use microorganisms, mainly bacteria, as catalysts to oxidize organic or inorganic matter and generate electricity. Electron acceptors in the cathode compartment play an important role in the performance of MFCs; oxygen and ferricyanide are the most commonly used ones, although recently the use of some pollutants with high electrochemical redox potentials that are present in wastewater as electron acceptors has been reported in MFCs while reducing at the same time [162,163,164]. An important aspect for the efficiency of the system lies in the design of a MFC; this is a key factor in the synthesis and production of electricity to obtain high efficiency and reduce costs for industrial applications. Some of the advantages of the MFC technology over other conventional systems are higher conversion efficiency than enzymatic fuel cells, sustainability of the process due to the efficiency of electricity conversion from the chemical energy of the substrate used [164,165], high efficiency in effluents with a low concentration of the organic pollutant load, low solids generation, adaptability to ambient temperatures, among other things [166,167,168]. Table 4 shows the different microorganisms used in MFC processes.

#### 3.4.3. Microalgae

Microalgae are autotrophic and photosynthetic aquatic microorganisms capable of synthesizing various high-value compounds, such as lipids, proteins, carbohydrates and pigments, which, together with a rapid growth rate and the ability to adapt to extreme environments, make them a viable option in the wastewater treatment [7,16,169,170]. This adaptive capacity allows them to grow under various conditions of acidic pH, high temperatures or excess of some compounds [171], allowing them to generate a greater amount of biomass and products of commercial interest [172,173]. Among the main pollutants in wastewater are organic compounds, nutrients and CO_2_; microalgae have the ability to assimilate some of these compounds for their metabolism (nitrogen, phosphorus, carbon, heavy metals, among other things), reducing the pollutant load of the water and minimizing eutrophication in water bodies [174], achieving efficient removal of up to 100% in some compounds. The main microalgae implemented in these treatment processes are *Chlorella* sp. [175], *Scenedesmus* sp. [176] and *Dunaliella salina* [177], among others. The ability of microalgae to use compounds present in wastewater reducing the pollutant load, strengthening the formation of metabolites and generating considerable amounts of oxygen that allow the proliferation of other microorganisms that contribute to the improvement of water quality makes them an interesting tool in wastewater treatment [178,179,180,181]. The adaptive capacity of microalgae, together with their ability to bioaccumulate heavy metals, allows them to be used as a potential system in the treatment of tannery wastewater. Table 5 shows different microalgae used in the treatment of tannery wastewater. It has been shown that this tolerant mechanism in *Chlorella* and *Scenedesmus* strains is due to the activation of antioxidant enzymes (superoxide dismutase, catalase and ascorbate peroxidase) as a biological response to oxidative damage induced by the presence of metals [182]. However, it is not yet possible to use the crude tannery effluent as a culture medium, which can be attributed to high toxic loads and the dark color of the crude effluent, which prevents the entry of light into the medium and limits the growth of microalgae, which suggests the application of wastewater dilutions to allow better adaptation [16,183] and determination of the appropriate photoperiod [7].

The contents of other compounds present in the wastewater are also influential in the efficiency of chromium removal since they compete for the interaction with the functional groups of the microalgae, showing that in synthetic wastewater with significant concentrations of chromium (20 mg/L), there is greater efficiency than in tannery wastewater, the interaction of pH, chromium concentration, strain used and temperature also playing an important role [43,184]. *Chlorella* strains have been extensively studied [182,184], exhibiting optimal growth in tannery wastewater diluted to 50%, guaranteeing the phycoremediation of heavy metals of up to 73.1%, 90.4%, 92.1 and 81.2% for Cr, Cu, Pb and Zn, respectively, contributing to the accumulation of a high yield of lipids (by 18.5%) and unsaturated fatty acids (by 50.05%) [185]. Likewise, studies with *C. vulgaris* also showed significant removals of DQO, NO_3_–N, PO_4_–P, SO_4_–S and Cr of 94.74%, 100%, 91.73%, 99% and 100%, respectively, between the 6th and the 21th days of culture [42]. In strains of *C. vulgaris* and *Pseudochlorella pringsheimii*, significant reductions in the concentration of contaminants were observed, greater than 65% for NH_3_–N, 100% for PO_4_–P, greater than 63% for COD and greater than 80% for total chromium, in dilutions of up to 30% of the residual water (6.25 mg/L of chromium); in addition, lipid accumulation of up to 25.4% was observed for *Pseudochlorella* and 9.3% for *Chlorella vulgaris* in a 20% dilution [43]. Da Fontoura et al. [186] reported positive results in strains of *Scenedesmus* using a central compound design, demonstrating that the concentration of wastewater and the intensity of light influence the amount of biomass produced and the removal of nitrogen and phosphorus. In this study, concentrations between 20% and 100% of the residual water were implemented, and removals of up to 85.631% were found for ammonia nitrogen, 96.78% for phosphorus and 80.33% for COD [13]. Ballén-Segura et al. [170] reported removal results for Cr(VI) greater than 98%, for nitrates—greater than 90%, for phosphates—greater than 99% and for BOD—greater than 88% (in similar concentrations). Regarding the concentration of heavy metals, efficiencies of 97%, 95%, 97% and 97% were found for Cr, Cu, Pb and Zn in dilutions of 10% of wastewater and of 57%, 79%, 48% and 65% for Cr, Cu, Pb and Zn, respectively, in the 100% residual water concentration. Other authors [2,12,30,170] have reported the ability of the genus *Scendesmus* to remove the organic pollutant load, COD—up to 97%, and Cr—up to 98%, present in tannery effluents. The genus *Tetraselmis* has also been reported in the treatment of tannery wastewater [7,16], reaching removals of 54% for COD, 97% for nitrogen compounds and 97% for phosphorus compounds.

**Table 5 molecules-26-03222-t005:** Algal strains used in the treatment of tannery wastewater.

Strain	Operating Conditions	Parameters	Removal Efficiency	Reference
*Scenedesmus* sp.	V: 1 L; *w*/*w* concentration: 20–100%; light intensity: 97.5–182.5 μmol photons m^−2^ s^−1^; pH: 7.5; T: 25 °C; time: 25 d.	COD NH_3_–N PO_4_–P	COD: 80.33% NH_3_–N: 85.63% PO_4_–P: 96.78%	[185]
*Scenedesmus* sp.	V: 3 L; pH: 2−11, T: 25–40 °C; dye concentration: 200−1500 mg L^−1^; contact time: 540 min.	Absorption of the AB–161 dye pH TOC Total nitrogen (TN)	AB–161: 69.83% TOC: 50.78% TN: 19.80%	[13]
*Nannochloropsis oculata*	V: 0.2 L; light intensity: 75 μmol photons m^−2^ s^−1^; photoperiod: 12:12; T: 25 °C; time: 15 d; pH: 7.6.	COD Color Inorganic carbon NH_4_–N PO_4_–P Chromium (Cr) TDS	COD: 84% Color: 60% Inorganic carbon: 90% NH_4_–N: 82% PO_4_–P: 100% Cr: 97% TDS: 10%	[43]
*Dunaliella salina*	V: 0.25 L; T: 25 °C; pH: 7.5; Cr (10, 20 and 30 mg L^−1^); culture temperature: 25 ± 2 °C (±1); photoperiod: 24:0; light intensity: 10 Wm^−2^; time: 120 h.	Cr	Cr: 66.4%	[177]
*S. quadricauda*	V: 1 L; photoperiod: 16:8 h (light/dark); light intensity: 110 μmol photons m^−2^ s^−1^; T: 22 °C; pH: 2–7; Cr concentration: 10 mg L^−1^; time: 8 d.	Cr(VI)	Cr: 98%	[12]
*Chlorella vulgaris* *Pseudochlorella pringsheimii*	V: 0.3 L; tanning effluents dilution: 10–50%; photoperiod: 24:0 h (light/dark); T: 27 °C; light intensity: 35 μmol photons m^−2^ s^−1^.	NH_3_–N PO_4_–P COD Cr	NH_3_–N: 100% PO_4_–P: 63% COD: 80% Cr: 56%	[43]
*C. pyrenoidosa* *Scenedesmus* sp.	V: 0.25 L; photoperiod: 12:12 h; T: 27 °C; tannery effluent concentration: 0−10–25–50–75−100%; pH: 7; time: 12 d.	Cr	Cr: 75%	[182]
*Tetraselmis* sp.	V: 0.25 L; T: 24 °C; photoperiod: 24:0 h (light/dark); time 19 d.; tannery effluent concentration: 50–75%.	Total Kjeldahl Nitrogen (TNK) NH_3_–N PO_4_–P Chemical oxygen demand (COD)	NH_3_–N: 99.90%, TKN: 79.36%, PO_4_–P: 87.82%, COD: 14.26%	[16]
*C. vulgaris* *S. acutus*	V: 10 L; T: 24 ± 2 °C; pH: 6.3 ± 0.3; time: 8−10 d.; illumination: 4500 ± 50 lux; photoperiod: 16:8 h (light/dark).	Cr	Cr: 88.2% (*C. vulgaris*), 87.1% (*S. acutus*)	[184]
*C. vulgaris*	V: 0.1 L; water concentration: 100–70–50–30−10%; T: 28 ± 0.5 °C; fluorescent lights: 150–300 μmol photons m^−2^ s^−1^; photoperiod: 10:14 h (light/dark); time: 21 days.	BOD COD NO_3_–N PO_4_–P SO_4_–S Cr	NO_3_–N: 100% Cr: 91.73% PO_4_: 99% SO_4_: 67.4% COD: 94.74% BOD: 95.93%	[42]
*Scenedesmus* sp. *C. variabilis* *C. sorokiniana*	V: 0.1 L; T: 25 °C, illumination: 40 μmol photons m^−2^ s^−1^; photoperiod: 14:10 h (light/dark); concentration: 25–40–60%.	COD NH_4_–N PO_4_^3−^–P	*Scenedesmus* sp.: COD: 66% and 56% NH_4_: 47% and 39% PO_4_^3−^: 70% and 64% *C. variabilis*: COD: 84% and 80% NH_4_: 68% and 62% PO_4_^3−^: 93% and 87% *C. sorokiniana*: COD: 80% and 74% NH_4_: 74% and 56% PO_4_^3−^: 93% and 93%	[2]
*Scenedesmus* sp.	V: 0.5 L; wastewater concentration: 20–50−100%; photoperiod: 16:8 h (light/dark); T: 24 °C; time: 15 d. The 100% concentration was used for nutrient removal experiments.	Cr(VI) NO_2_–N NO_3_–N PO_4_–P SO_4_–S DBO	Cr^+6^: 98% NO_2_: 95% NO_3_: 90% PO_4_: 99% SO_4_: 92% BOD_5_: 88%	[170]
Microalgae consortium. Dominant microalgae: *Tetraselmis* sp.	V: 0.25 L; wastewater concentration: 50R50S and 75R25S; photoperiod: 12:12 h (light/dark); air flow: 1 L min^−1^; time: 20 days.	PO_4_–P Total nitrogen (TN) NH_3_–N COD TOC BOD	50R50S: PO_4_: 97.6%, TN: 71.7%, NH_3_: 100%, COD: 50.4%, TOC: 20%, BOD_5_: 16.8% 75R25S:PO_4_: 95.5%, TN: 58.8%, NH_3_: 100%, COD: 56.7%, TOC: 31.3%, BOD_5_: 20.7%	[7]
*Scenedesmus* sp.	V: 0.15 L; tannery wastewater dilutions: 10%, 25%, 50%, 75% and 100%; T: 27 ± 2 °C; illumination: 4000 lux; photoperiod: 16:8 h (dark/light); time: 12 d.	Cr Cu Pb Zn NO_3_ PO_4_	Cr: 81.2–96% Cu: 73.2–98% Pb: 75–98% Zn: 65–98% NO_3_ > 44.3% PO_4_ > 95%	[30]
*C. vulgaris*	V: 0.25 L; T: 26 ± 2 °C; illumination: 5000 lux; wastewater concentration: 100%; pH: 7.1; time: 15 d.	NO_3_–N NH_4_–N PO_4_–P COD	NH_4_: 55% NO_3_: 85.6% PO_4_: 60.5% COD: 43.4%	[186]
*Chlorella* sp.	V: 0.3 L; wastewater tannery concentration: 50−100%; time: 12 d; T: 27.5 °C; illumination: 4000 lux; photoperiod: 12:12 h (fluorescent lamps).	Cr Cu Pb Zn	50% dilution: Cr: 73.1%, 45.7% Cu: 90.4%, 78.1% Pb: 92.1%, 52.2% Zn: 81.2%, 44.6% 100% wastewater: Cr: 45.7% Cu: 78.1% Pb: 52.2% Zn: 44.6%	[186]
*Chlorella* sp. *Phormidium* sp.	V: 15 L; tannery wastewater: 100%; time: 20 d; T: 28 °C; light intensity: 225 μmol photons m^−2^ s^−1^; photoperiod: 12:12 h	BOD COD TN Total phosphorus (TP) Cr TDS	BOD: 93.4% COD: 96.6 ± 11.1% TN: 91.16% TP: 88% Cr: 94.45% TDS: 58.28%	[173]

## 4. Present and Future Prospects

Currently, the implementation of composite systems between advanced oxidation and phytoremediation processes are being evaluated (Figure 4). Studies have shown an optimal correlation between ozonation and phytoremediation with the microalgae *Nannochloropsis oculata*, where a removal of 84% was achieved for COD, 60% for color, 100% for odor, 90% for inorganic carbon, 84% for NH_4_^+^–N, 100% for PO_4_–P, 97% for chromium and 10% for TDS; however, it is necessary to maximize the efficiency of ozone utilization to minimize operating costs and continue research regarding operating conditions [43]. Microalgae have demonstrated their ability to absorb various heavy metals, exhibiting greater effectiveness under certain conditions, as in the case of *Dunaliella salina*, which exhibited a greater capacity for chromium (VI) biosorption with an efficiency of 66.4% at an optimal pH (8.6) and an inoculum size of 10% within the first 120 h, demonstrating a viable solution for the bioremediation of wastewater with a high content of heavy metals [177].

The above demonstrates the capacity of microalgae to reduce most of the highly environmentally polluting compounds in tannery wastewater, which would allow it to be an ecological strategy through the establishment of bioremediation centers and economic biorefineries to recover resources, guaranteeing the recovery of chromium for reuse in the tanning process and promoting sustainable economic development [185,187]. The use of microalgal biomass resulting from the biomeremediation process of tannery effluents to obtain biofuels has been reported due to the accumulation of lipids [188] being the main use; the success of the commercial application of these biofuels is associated with the biorefining of other metabolites such as pigments or renewable polymers that can be used [189]. Finally, the path to commercialization of the microalgae fuel is related to the implementation of improvements in metabolic engineering, synthetic biology and genomics, the development of closed photobioreactor systems and the invention of new lighting, harvesting, and extraction systems [188]. Advanced oxidation processes and biological processes have gained popularity in recent years; however, they require further study to specify the optimal conditions for their operation. Therefore, one of the prospects in the tannery wastewater treatment technology is the development of various investigations that promote the combined use of AOPs and the microalgae biotechnology, where operating conditions are evaluated and variables are determined, that allow improving productivity of the biomass and metabolites of interest with a potential in various sectors of the industry.

## 5. Conclusions

In recent years, different strategies have been implemented for the treatment of tannery wastewater, as this process leads to the generation of polluting loads with high environmental impact. However, the main treatment systems, although efficient, can become expensive, difficult to operate and at the same time generate secondary waste that turns out to be more toxic than the raw wastewater would be. For this reason, over the years, it has been sought to design strategies that allow easy access and are efficient in removing the pollutant load.

Advanced oxidation processes and biological processes have gained popularity in the recent years; however, they require further study to specify the optimal conditions for their operation. Among the main biological processes for the treatment of tannery waters is phycoremediation with microalgae since their adaptive capacity to stress situations allows them to develop adequately in this type of water. Therefore, various investigations have been developed that promote its use, finding high removal efficiencies and the implementation of combined systems. Despite this, further studies are required to determine the adaptive capacity of various strains since currently the most widely used are *Scenedesmus* sp. and *Chlorella* sp. Among the main advantages of the implementation of microalgae in tannery wastewater is the reuse of nitrogen and phosphorus remaining from the tanning process as nutrients for its metabolism, its ability to generate metabolites of industrial interest obtained due to the stress developed during growth in this medium and the bioabsorption capacity of chromium for its subsequent recovery. For these reasons, phytoremediation with microalgae coupled to advanced oxidation processes such as tannery wastewater treatment promises to be a viable strategy, being an affordable, ecosustainable process, with efficient removal and better productivity of the microalgal biomass.

## Figures and Tables

**Figure 1 molecules-26-03222-f001:**
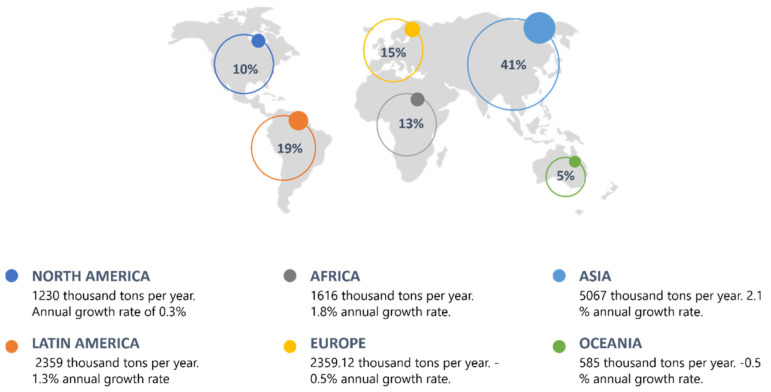
Global context of the leather industry.

**Figure 2 molecules-26-03222-f002:**
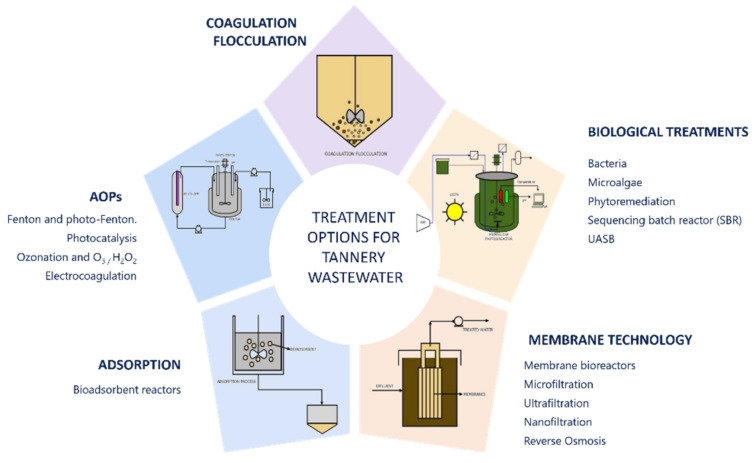
Treatment options for tannery wastewater.

**Figure 3 molecules-26-03222-f003:**
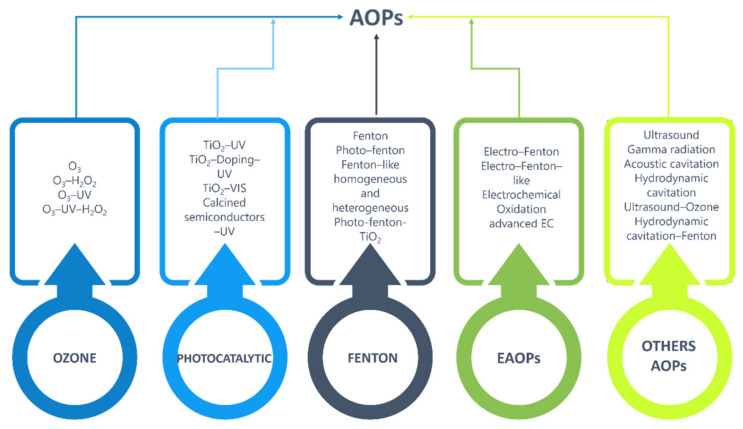
Types of oxidation processes used in the treatment of tannery wastewater.

**Figure 4 molecules-26-03222-f004:**
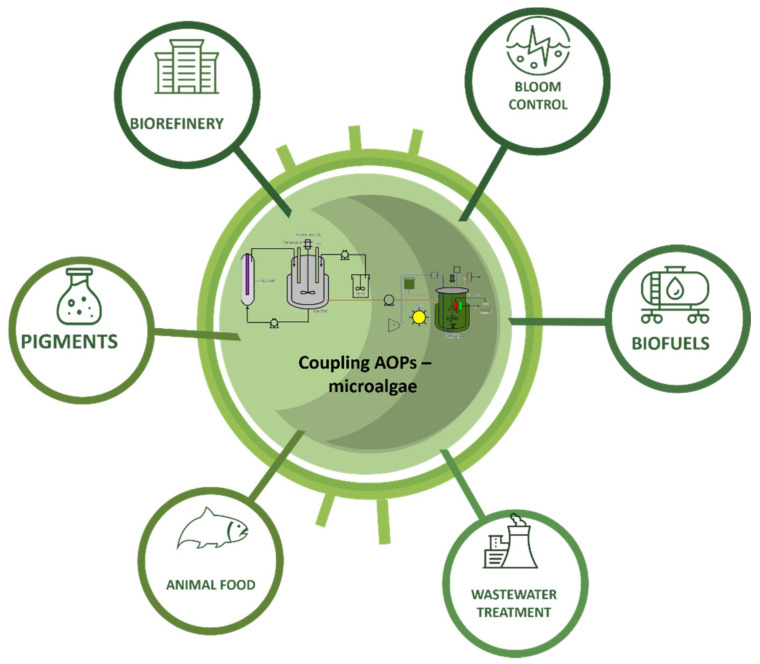
Biotechnological applications of the coupling of microalgae and AOPs.

**Table 1 molecules-26-03222-t001:** Physicochemical characterization of different tannery wastewater.

pH	COD (mg*L^−1^)	BOD (mg*L^−1^)	TDS (mg*L^−1^)	Cr (mg*L^−1^)	NH_3_–N (mg*L^−1^)	PO_4_ (mg*L^−1^)	Reference
n/a	17,683 ± 1500	6000 ± 300	10,000 ± 800	n/a	4500	4100	[9]
7.5	4000	1400	n/a	n/a	343	6.6	[13]
3.4–3.7	5250–9600	n/a	38,200–39,400	2705–3800	115–136	n/a	[35]
3.5–4	6800	n/a	n/a	n/a	n/a	1.76	[36]
8.49 ± 0.2	322 ± 28.6	160 ± 15.8	3491.3 ± 239.4	1445 ± 67.9	n/a	5.7 ± 0.2	[37]
7.9	4155	–	n/a	n/a	485	524	[38]
8.9 ± 0.1	4500 ± 329	400 ± 36	5900 ± 391	n/a	129.69 ± 7.75	194.61 ± 9.8	[39]
5.84 ± 0.02	198.60 ± 0.23	6.5 ± 0.10	72,400 ± 0.10	0.83 ± 0.28	n/a	294.4 ± 0.22	[40]
8.45 ± 0.18	1300.00 ± 10.0	680.00 ± 20.0	3850.00 ± 10.0	7.39 ± 0.03	12.3 ± 0.3	12.5 ± 0.5	[41]
7.45 ± 0.00	4000.00 ± 51.20	n/a	4333.33 ± 288.70	3.22	2734.16 ± 1.12	6.01 ± 0.05	[42]
4.13	5485	90	n/a	2007.08	n/a	n/a	[44]
4–9	1235	450	n/a	128.8	n/a	n/a	[45]
11.64 ± 0.53	7200 ± 1090	1250 ± 380	n/a	7.02 ± 0.76	n/a	n/a	[46]
6.25	11,800	1200	n/a	32.2	n/a	n/a	[48]
8.36	5308.4	1952.5	1578	123.1897	n/a	n/a	[49]
7.5	4291 mg/L	2102.60	n/a	n/a	n/a	n/a	[50]
3.78	1980	n/a	n/a	3060	n/a	n/a	[56]
8.67 ± 3.5	7273 ± 536	3120.6 ± 172	n/a	28.47 ± 5	112.2 ± 24	n/a	[51]
11.3 ± 0.1	3000 ± 100	n/a	n/a	50 ± 3	n/a	n/a	[52]
3.9 ± 0.1	4321 ± 21.2	3200 ± 77	42,200 ± 100	2920.2 ± 0.7	n/a	n/a	[53]
8.0 ± 0.4	5634 ± 245	2910 ± 341	10,560 ± 978	134 ± 5.8	n/a	n/a	[54]
8.6 ± 0.1	12,560 ± 1880	4860 ± 129	18,250 ± 1825	n/a	n/a	n/a	[55]
3.17	1130	n/a	n/a	1640	n/a	n/a	[57]
6.5	2530	n/a	822	57	57	n/a	[58]
8.8	2780	1225	n/a	8.2	n/a	n/a	[59]
9.3−12.1	1500 ± 400	n/a	n/a	360 ± 110	n/a	n/a	[60]
8.7 ± 0.2	2412 ± 145	649.3 ± 39.3	2355 ± 85	8.11 ± 4.86	n/a	n/a	[61]
9	17,600	n/a	6900	120	n/a	916	[62]
4.0 ± 0.12	300 ± 2.08	250 ± 1.62	19.426 ± 3.06	25	n/a	n/a	[63]
6.85	987	580	1185.4	12.4	n/a	n/a	[64]
4.12	3280	n/a	n/a	147.4	n/a	n/a	[74]

**Table 2 molecules-26-03222-t002:** AOP in tannery wastewater.

Process	Operating Conditions	Evaluated Parameters	Efficiency	Reference
Cavitation	The amount of energy dissipated in 250 mL was 0.122 W*mL^−1^	COD	87%	[50]
Fenton	V: 50 mL, T: 25 ± 0.1 °C, agitation: 150 rpm, FeSO_4_: 1–5 g L^−1^, time: 5–300 min; H_2_O_2_/COD ratio (*w*/*w*): 0.5–1.0.	COD	58.4%	[95]
	V: 500 mL, pH: 3, T: 40–45 °C, H_2_O_2_: 0.15–0.6 g L^−1^, FeSO_4_: 500–750 mg L^−1^, time: 0–30 min.	COD	68%	[96]
	3V: 300 mL, agitation: 150 rpm, time: 60 min; Fe^2+^ dosage: 0–20 mg L^−1^, pH: 3–7, H_2_O_2_ dosage: 50–100 mg L^−1^.	COD Color Turbidity Sludge	COD: 80% Color: 90% Turbidity: 95% Sludge: 70%	[98]
Fenton + NaOCl and Fenton + (NH_4_)_2_S_2_O_8_	V: 100 mL, pH: 3.5, agitation: 200 rpm, Fe^2+^ dosage: 11.5 mg/g DS, H_2_O_2_ dosage: 167.0 mg/g DS, time: 12 min.	Cr	73.3%	[97]
Photo-Fenton	V: 500 mL, solar irradiation: 5 h; Fe^2+^: 0.4–0.5 g L^−1^; H_2_O_2_: 15–30 g L^−1^, pH: 3, time: 2 h.	COD TDS	COD: 90% TSS: 50%	[100]
Ozone	V: 2500 L, flow rate: 2 m^3^ h^−1^; O_3_ dosage: 150 g m^−3^, time: 60 min, pH: 6.8.	COD TSS TKN Color	COD: 97% TSS: 96% TKN: 91% Color: 96%	[101]
	V: 5 L, pH: 4–7–9, O_3_ dosage: 1.6 mg L^−1^, time: 10–20–30–40–50 min.	Color	97% in a time of 20 min and a pH of 7	[102]
	V: 3 L, pH: 3–6–9, Ozone flow rate: 1 and 8 g h ^−1^. time: 10–20–40–60–90−120 min, T: 27 °C	COD	COD: 70%	[104]
Fenton and Ozone	V: 0.5 L, Fe^2+^ concentration: 120 to 300 mg L^−1^, concentration of H_2_O_2_: 600–2000 mg L^−1^, pH: 4, Ozone flow: 1 L min^−1^.	COD	COD: 60–70%	[105]
Ozone coupled with phycoremediation	V: 1 L, pH: 3.7–6–9, ozone flow rate: 2–4–6 g h^−1^, time: 10–20–40–60–90−120 min.	COD, сolor, Cr, NH_4_, PO_4_, TDS	COD: 84% Color: 60% Cr: 97% NH_4_–N: 82% PO_4_–P: 100% TDS: 10%	[43]
Electrochemical	V: 2 L, total surface area: 427.84 cm^2^; pH: 3–9; salt concentration: 10–40 g L^−1^ NaCl; time: 120 min.	COD	COD: 89% to 0.012 A cm^−2^ pH: 9, salt concentration: 30 g*L^−1^ NaCl	[107]
	V: 1.15 L, total surface area: 69.75 cm^2^, pH: 2−11; current density: 3.5–70 A cm^−2^; time: 10–70 min.	OD	COD: 62% in a range of pH 3–5 and time: 10 min	[83]
Electrochemical/photo-Fenton/Fenton	V: 1.5 L, pH 8.3; current density: 68 mA cm^−2^; currents and voltages: 0–10 A and 0–30 V; t: 5–60 min.	COD, color, turbidity	COD: 99% Turbidity: 98% TSS: 65%	[103]
	V: 4 L, pH: 3; anode and cathode electrode area: 64 cm^2^; time: 180 min.	COD, color	COD: 90% Color: 86%	[108]
	V: 500 mL, pH: 3.0, H_2_O_2_ concentration: 0.5 Mm; Fe^2+^ concentration: 0.50 mM.	COD, color	Color: 97% COD: 95%	[109]
Electrocoagulation combined with photoreactor UVC	V: 0.2 L, UV lamp wavelength: 254 nm and 185 nm; electric current: 100–600 mA; time: 10–30 min.	COD, Cr	COD: 99.52% Cr: 98.27%	[110]
Photocatalysis	Air flow: 140 N cm^3^ min^−1^; four UV lamps: power: 8 W, wavelength: 350 nm; photon flux: 25 mW/cm^2^.	COD	The ZnO_ac1 photocatalyst achieved a COD removal of 70% in 180 min of irradiation	[111]
	V: 5 L; without pH adjustment; time: 5 h in PTR; exposed directly to sunlight.	COD, Cr	COD: 82.26% Cr: 76.48%	[112]

**Table 3 molecules-26-03222-t003:** Energy consumption and costs of different AOPs.

AOP Type	EEO (kWh m^−3^ Order^−1^)	EEM (kWh g^−1^)	Cost (US$ m^−3^)	References
O_3_	0.3	495	11.3	[114,122,124,125,126,127,128,129,130]
O_3_/H_2_O_2_	0.2	-	8.6	
UV/O_3_	225.25	111.56	6	
Photo-Fenton	12	-	64.13	
Photoelectro-Fenton	132.6	0.125	8.4–66.22	
Electro-Fenton (EF)	127.2	0.235	8.48	
UV/US/H_2_O_2_	39.76	167	4.49	
Ultrasound	800–8000	11,993	55.14	
Photocatalysis	3654.68	21,129.15	2.8	
Electrocoagulation	59.4	0.060	3.94	

**Table 4 molecules-26-03222-t004:** Different microorganisms used in MFC processes.

Microorganism	Removal Efficiency	MFC Performance	Reference
*Bacillus* sp.	COD: 88%	120 mA/m^2^ and 7 mW/m^2^	[155]
Anaerobic microbial consortium	COD: 48.5%	44.2 and 52.1 mW/m^2^	[156]
*Chlorococcum* sp. *Synechococcus* sp.	COD: 73.5% (*Chlorococcum* sp.) 69.4% (*Synechococcus* sp.)	*Chlorococcum* sp.: 32.1 ± 0.5 and 27.2 ± 0.5 mW/m^2^ *Synechococcus* sp.: 42.5 ± 0.5 and 37.2 ± 0.3 mW/m^2^	[157]
Activated sludge consortium	NO_3_^−^: 87% COD: 90%	0.35 mA·cm^−2^ and power level of 6.11 mW	[159]
Anaerobic sludge	COD: 98%	88 mW/m^2^ and 408 mA*m^−2^	[160]
*Shewanella decolorationis* S12, *Klebsiella pneumoniae* L17	COD: 42.5%	52.1 mW/cm^2^ with an air bubbling cathode 6.8 mW/cm^2^ with a nitrogen bubbling cathode	[161]
Algae biomass	COD: 72–95% Cr: 98%	221 mV to 760 mV	[162]
Anaerobic microbial consortium	Cr^6+^: 95%	89 ± 3 mW/m^2^	[163]
Adapted microbial consortium	Cr: 71.4%	970.2 ± 20.6 mW/m^2^	[167]
*Trichococcus pasteurii* *Pseudomonas aeruginosa*	COD: 98%	55.5 mW/m^2^	[168]

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
