# Peer review of "Advanced Oxidation Processes and Biotechnological Alternatives for the Treatment of Tannery Wastewater"

_molecules, 2021, doi:10.3390/molecules26113222_

Round 1

Reviewer 1 Report

The manuscript entitled “Advanced oxidation processes and biotechnological alternatives for the treatment of tannery wastewater” shows interesting issues related to the treatment of tannery wastewaters by advanced oxidation technologies and biological treatments. However, the issue about advanced oxidation processes is too limited and it requires more information about why AOPs seem to be a promising technology for the treatment of these residues. Moreover, I suggest that the manuscript needs to have a more critical view of the limitations and perspectives of mentioned technologies. I also suggest enhancing English grammar. My decision is accepted with major changes.  

Line 49: I suggest removing the information about Cundinamarca (Colombia) and adding information about what is the position of Colombia in the global and Latin American leather market.

Line 55: It would be interesting if authors mention how much wastewater is produced by tannery industries.

Line 101: Please remove data concerning local aspects of the Colombia tannery industry. I would leave the information about what kind of microorganisms have been used in Colombia to treat tannery wastewaters.

Line 158: change soluble by solubility.

Line 226: Please change the sentence potent free radical molecules by “highly oxidative radicals”. Hydroxyl radical must have a point in the O atom. Remember that AOPs generate mainly hydroxyl radicals, so, I would focus the interest on this radical species.

Line 242: In the sentence “Another advanced oxidation method is the use of ozone as the most powerful oxidant, implemented in wastewater treatment thanks to its ability to reduce color, synthetic aromatic compounds and persistent organic pollutants (POPs)” what did mean authors with most powerful oxidant? Ozone (2.07 V vs. NHE) is not a high oxidant species compared with hydroxyl radicals (2.86 V vs. NHE).

Line 247: The sentence “A technology that has had great relevance in recent years in advanced oxidation methods are advanced oxidation processes to electrochemical [83,103],…” is not clear. Please revise it.

Line 281: I suggest writing beside Bogotá, Colombia in brackets.

Line 319: Please revise the sentence: “the microorganisms present generate electrons in the anode chamber and then these electrons are transported to the cathode chamber where oxygen [141–142].”

Author Response

The manuscript entitled “Advanced oxidation processes and biotechnological alternatives for the treatment of tannery wastewater” shows interesting issues related to the treatment of tannery wastewaters by advanced oxidation technologies and biological treatments. However, the issue about advanced oxidation processes is too limited and it requires more information about why AOPs seem to be a promising technology for the treatment of these residues. Moreover, I suggest that the manuscript needs to have a more critical view of the limitations and perspectives of mentioned technologies. I also suggest enhancing English grammar. My decision is accepted with major changes. 

Answer: The authors thank the comments and suggestions. They were considered for this new version. The red color was used in the text to highlight the answer to the questions

Line 49: I suggest removing the information about Cundinamarca (Colombia) and adding information about what is the position of Colombia in the global and Latin American leather market.

Answer: the changes about Cundinamarca region was delete and the global position of Colombia about market was included

Line 55: It would be interesting if authors mention how much wastewater is produced by tannery industries.

Answer: the data (8 to 20 m3 of wastewater) was included into the introduction section

Line 101: Please remove data concerning local aspects of the Colombia tannery industry. I would leave the information about what kind of microorganisms have been used in Colombia to treat tannery wastewaters.

Answer: The local aspects were deleted and the new information about microorganisms was included. In Colombia, several research studies have been carried out to mitigate the impacts, among them are the identification and evaluation of pollutants [31, the use of the Eichhornia crassipes plant has been reported in pilot-scale treatment of tannery water due to its capacity to accumulate heavy metals and organic matter [32], the genotoxic effect of tannery effluents on Salmonella typhimurium strains has also been studied, identifying a mutagenic increase and the capacity to generate DNA damage in human lymphocytes [31- 32]. However, the application of microalgae in tannery wastewater has not yet been developed in depth in this country.

Line 158: change soluble by solubility.

Answer: it was made.

Line 226: Please change the sentence potent free radical molecules by “highly oxidative radicals”. Hydroxyl radical must have a point in the O atom. Remember that AOPs generate mainly hydroxyl radicals, so, I would focus the interest on this radical species.

Answer: it was checked and modified.

Line 242: In the sentence “Another advanced oxidation method is the use of ozone as the most powerful oxidant, implemented in wastewater treatment thanks to its ability to reduce color, synthetic aromatic compounds and persistent organic pollutants (POPs)” what did mean authors with most powerful oxidant? Ozone (2.07 V vs. NHE) is not a high oxidant species compared with hydroxyl radicals (2.86 V vs. NHE).

Answer: the sentence was modified. Another advanced oxidation method is ozone, which has been implemented in wastewater treatment due to its ability to reduce color, synthetic aromatic compounds and persistent organic pollutants (POPs)

Line 247: The sentence “A technology that has had great relevance in recent years in advanced oxidation methods are advanced oxidation processes to electrochemical [83,103],…” is not clear. Please revise it.

Answer: The sentence was modified. Electrochemical processes have had great relevance in recent years [83,103], this technology generates oxidizing agents, destroying organic compounds until their mineralization, using methods of anodic oxidation, photoelectrocatalysis [104,105] and electro-phenon

Line 281: I suggest writing beside Bogotá, Colombia in brackets.

Answer: it was made.

Line 319: Please revise the sentence: “the microorganisms present generate electrons in the anode chamber and then these electrons are transported to the cathode chamber where oxygen [141–142].”

Answer: This was checked and modified.

MFCs are devices that use microorganisms, mainly bacteria, as catalysts to oxidize organic or inorganic matter and generate electricity. Electron acceptors in the cathode compartment play an important role in the performance of MFCs, oxygen and ferricyanide are the most commonly used ones used, although recently it has been reported the use of some pollutants with high electrochemical redox potentials that are present in wastewater, as electron acceptors in MFCs, while reducing at the same time

Reviewer 2 Report

This manuscript present a literature overview on the advanced oxidation processes and biotechnological alternatives for the treatment of tannery waste water. The authors should consider the below comments in order to improve their paper.

  1. The authors state in the abstract that “Europe and Asia are the main producers of this industry” but according to figure 1 Europe is overpassed by Latin America and Africa. It should be clarified.
  2. Page 2, line 55: is between 25-40 m3 or 10-100 m3? because it is confusing.
  3. Page 2, line 63: chromium is not a compound it is a metal.
  4. Please revise section 2 because it is overlapping with table 1 and there are domains which are in contradiction between the text and table 1 (e.g. Cr content).
  5. I do not see the relevance of section 3 because it was described sufficiently clear in the introduction section. In my opinion it should be deleted considering that it does not provide any novel information besides the introduction section. The same is true for section 4.
  6. It would be great to compare the technological options based on the specific energy consumption for the treatment of a specific amount of wastewater. This would be a novel approach to present and compare different tannery waste water treatment alternatives.
  7. The section about coagulation does have no importance considering that it is not an AOPs and neither biochemical one. Moreover, it is too short to provide a comprehensive literature overview.
  8. Regarding the electrochemical processes it would be very useful if the manuscript would provide data regarding the amount of Cr electrodeposited in comparison to the total amount removed. This way the readers should be informed by the potential use of electrochemical processes for the production of Cr in waste treatment technologies.
  9. The section about AOPs is to short in comparison to section 5.4. Please provide a more detailed presentation considering the many possible alternatives shown in figure 3. In my opinion the data from table 2 should be discussed in more detail and different AOPs need to be compared to each other in order to provide a comprehensive overview of AOPs.
  10. Page 8, line 248 “advanced oxidation processes to electrochemical” this term is not clearly defined.. what does this mean? Please clarify it.
  11. Page 11, lines 262-263 “different microorganisms have been proposed as an efficient and economical alternative”. Please provide data to sustain this affirmation. I do not see any values which confirm that it is an economical alternative.
  12. Page 13, line 319 “...transported to the cathode chamber where oxygen..”. This sentence is not complete.
  13. Please delete mW/m2 from the head of table 3 and % from the values in column 2.
  14. Page 15, lines 410-411..what about the accumulated heavy metals in the formed biomass. What is their environmental impact considering that they are concentrated in the biomass?
  15. Please insert a list of abbreviations. Also, the unit for the involved parameters needs to be added in accordance with the SI.

Author Response

This manuscript present a literature overview on the advanced oxidation processes and biotechnological alternatives for the treatment of tannery waste water. The authors should consider the below comments in order to improve their paper.

Answer: The authors thank the comments and suggestions. They were considered for this new version. The blue color was used in the text to highlight the answer to the questions

  1. The authors state in the abstract that “Europe and Asia are the main producers of this industry” but according to figure 1 Europe is overpassed by Latin America and Africa. It should be clarified.

Answer: the information was checked and modified.

  1. Page 2, line 55: is between 25-40 m3 or 10-100 m3? because it is confusing.

Answer: the data was checked and modified

  1. Page 2, line 63: chromium is not a compound it is a metal.

Answer: the term was corrected.

  1. Please revise section 2 because it is overlapping with table 1 and there are domains which are in contradiction between the text and table 1 (e.g. Cr content).

Answer: the section 2 was revised and modified according to recommendations.

Different studies have reported high values of BOD, COD and even the presence of Cr in effluents [13, 41, 44, 44, 50, 55, 56, 58, 60, 64, 65, 67]. Other compounds have been reported in these wastewaters, finding results of acidic pH between 3.4 ± 0.0351 to 5.96 ± 0.0351 [47, 54, 57] and basic pH between 8.0 ± 0.4 to 11.64 ± 0. 53 [ 45, 46, 48, 52, 53, 53, 58, 60, 63, 64, 67] , in relation to TDS typical values can have concentrations ranging from 2355 ± 85 mg* L-1 to 10000 ± 800 mg* L-1 [9,46,50,53,66,68], high concentrations have been recorded that can be between 10560 ± 78 mg* L-1 to 72400 ± 0. 10 mg*L-1 [40,58,59,61,65,67]; as for BOD the average values can range in low ranges from 160 ± 15.8 mg* L-1 to 1250 ± 38 mg* L-1 [46, 50, 53, 61] and high ranges that oscillate between 1500 ± 41 to 6000 ± 30 mg* L-1 [9, 55, 56, 58, 60, 65, 67] . In relation to total chromium concentration, values ranging from 0, 83 ± 0.028 mg* L-1 to 134 ± 5.8 mg* L-1 have been reported [9, 40, 49, 59, 59, 62, 63] and high ranges from 147. 4 ± 1.5 to 3800 ± 115 mg* L-1 [40,42,46,52,54,57,65],

  1. I do not see the relevance of section 3 because it was described sufficiently clear in the introduction section. In my opinion it should be deleted considering that it does not provide any novel information besides the introduction section. The same is true for section 4.

Answer: the section 3 and 4 was modified according to the comments of reviewers

  1. It would be great to compare the technological options based on the specific energy consumption for the treatment of a specific amount of wastewater. This would be a novel approach to present and compare different tannery waste water treatment alternatives.

Answer: a new section was included for comparison of AOPs

3.3.1 Energy and cost considerations in AOPs

The selection of an AOP for wastewater treatment is not only based on the removal efficiency, although it is important, the energy consumption or efficiency of an AOP has taken great relevance when selecting a process. The electrical energy per order (EEO) is an indicator used to determine energy efficiency [122], defined as the amount of kWh required to reduce the concentration of a pollutant by an order of magnitude in a given volume of treated wastewater (kWh m-3 log-1) [123]. This parameter has been used to analyze and compare different AOPs in the treatment of pollutants present in wastewater, the reported OEE values are in the range of 0. 1 - 15 kWh m-3 log-1 for processes such as: ozonation, UV/O3, UV/H2O2, EB, Photo-Fenton, O3/H2O2; some processes such as: UV/catalyst, Microwave and Ultrasound report high values in the range of 150 - 8000 kWh m-3 log-1 [124], the OEE value varies in relation to the process efficiency, the higher the process efficiency the lower the calculated EEO value. 

Another aspect that affects the EEO value is related to the characteristics of the wastewater and the type of pollutants present, a very complex wastewater matrix with several pollutants will obtain high EEO values compared to simple matrices or with only one pollutant [121]. Wardenier et al., [126] conducted an investigation comparing UV,O3 and H2O2 efficiency for the treatment of micropollutants such as atrazine, alachlor, bisphenol, and found that energy consumption and treatment cost were in the order of: UV> UV/ H2O2 > UV/O3 > UV/O3/H2O2 > O3/H2O2 > O3, while the order in relation to removal efficiency was O3/H2O2 > UV/O3/H2O2 > UV/O3 > UV/ H2O2 > UV/ H2O2 > O3 > UV. In an azo dye treatment process implemented ultrasound with AOPs to achieve 90% removal, the order in relation to cost and energy consumption was: US > Photocatalysis > UV > UV/H2O2 > UV/O3 > Photo-Fenton> O3; in relation to percentage removal the order was:  Photocatalysis>Photo-Fenton>UV/H2O2>O3 [126]. Performing these calculations and comparing by technology provides information for process scale-up as well as for sustainability analysis. Table 3. shows a comparison in terms of energy consumption and cost of different AOPs.

Table 3. Energy consumption and costs of different AOPs.

AOP Type

EEO

 (kWh m−3 Order−1)

EEM

(kWh g-1)

Cost

(US$ m-3)

REFERENCES

 O3

0.3

495

11.3

[110,122,124-125,127-131]

O3/H2O2

0.2

-

8.6

UV/O3

225.25

111.56

6

Photo-Fenton

12

-

64.13

Photoelectro-Fenton

132.6

0.125

8.4-66.22

Electro-Fenton (EF)

127.2

0.235

8.48

UV/US/H2O2

39.76

167

4.49

Ultrasound

800-8000

11993

55.14

Photocatalysis

3654.68

21129.15

2.8

Electrocoagulation

59.4

0.060

3.94

  1. The section about coagulation does have no importance considering that it is not an AOPs and neither biochemical one. Moreover, it is too short to provide a comprehensive literature overview.

Answer: it was revised and modified according to recommendation. The data was included.

In relation to the removal of Cr (VI), electrocoagulation has been implemented in recent years, in this process flocculation occurs "in situ" thanks to the electrooxidation of a sacrificial anode (usually Fe or Al) [112]. One of the advantages of this process compared to chemical coagulation lies in the sludge production, the sludge reduction in electrocoagulation is 50% compared to chemical coagulation, showing more environmentally friendly properties. The process consists of transforming (directly or indirectly) Cr (VI) into Cr (III) and then precipitating and separating Cr (III) as hydroxide [54]. Removal efficiency of Cr (VI) has been reported up to 99% in a pH range of 5 and 8, above this range the removal efficiency decreases up to 27% while at pH lower than 5, 50% of Cr remains dissolved and the rest is electrodeposited in the sludge generated, therefore the control of the pH of the solution is a variable that affects the process.  It is known that chromium deposition is possible from solutions based on much less harmful Cr(III) compounds, which are obtained by electrochemical processes. These electrolytes can be a real alternative, however, the electrochemical reactions that take place in the electrodeposition of chromium from Cr(III) salt solutions are complicated and their understanding still needs to be studied. In an electrodeposition process using electroplating, the removal of chromium ions from tannery wastewater was evaluated. from a synthetic trivalent chromium solution and removed 96.5% of the total chromium content in the untreated effluent [113]. Finally the route dictated by the thermodynamics of the multi-step reduction of Cr(III) to Cr evidences that metallic chromium is probably deposited through the discharge of electroactive hydroxocomplexes of bivalent chromium that form in the near-cathode layer due to the dissociation of water molecules [114, 115] but it is still necessary to evaluate this process more extensively in tannery wastewater.

  1. Regarding the electrochemical processes it would be very useful if the manuscript would provide data regarding the amount of Cr electrodeposited in comparison to the total amount removed. This way the readers should be informed by the potential use of electrochemical processes for the production of Cr in waste treatment technologies.

Answer: The data was included.

  1. The section about AOPs is to short in comparison to section 5.4. Please provide a more detailed presentation considering the many possible alternatives shown in figure 3. In my opinion the data from table 2 should be discussed in more detail and different AOPs need to be compared to each other in order to provide a comprehensive overview of AOPs.

Answer: The discussion in this section was improved.

In AOPs the reaction of OH and the various pollutants present in tannery effluents results in mineral end products that produce inorganic ions and carbon dioxide [110]. The efficiency of each process depends on the physicochemical characteristics of the pollutants present in the tannery effluents, as well as on the generation of hydroxyl radicals. The generation of these radicals can be achieved by different processes, ozone, is a technology that has been reported in the treatment of dyeing water, reaching removals of 90-98% COD and color of 96% [ 97-98], the efficiency of the process depends largely on the pH, at acid values < 4. 5, the reaction is direct, molecular ozone dominates the reaction being selective mainly in the destruction of chromophore groups, while at pH >7 ozone decomposes generating OH which is less selective and has a higher oxidation potential; Ozone flow rate is another important variable, the percentage of removal is directly proportional to the ozone flow rate with respect to time, increasing the ozone rate increases the removal efficiency, it has been reported that excess ozone can lead to the formation of residual H2O2, allowing the wastewater concentration to be increased so that the COD present can be degraded by the excess H2O2 [110].

In relation to photocatalytic processes, photo-Fenton is one of the most studied in tannery effluents, and has achieved removals of 70-90 % COD, 86-98 % color [96, 104], and 90 % Cr [110], the efficiency of the process depends largely on the pH of the solution, the optimal range of higher catalytic activity is 2.8-3. 0, pH values > 5 generate ferric hydroxides that reduce the reactivity of OH, while at values below 2 complex iron species are formed that react more slowly with H2O2 decreasing the efficiency of the process [116]; the amount of ferrous ions generated also affects the process, excess concentrations in the solution can generate precipitates increasing the TDS concentration [117]. Finally, the irradiation time is another variable that affects the process, this should be as low as possible to minimize energy consumption without affecting process efficiency. The UV / H2O2 system has great relevance in the treatment of tannery effluents, the main reason being the absence of sludge production and significant COD removal in very short reaction times [118].

The effectiveness of the UV / H2O2 process for the degradation of complex compounds present in these effluents depends on several factors, the pH affects the reactivity of H2O2 as well as the generation of the OH radical, therefore, a pH of 3-5 is recommended to implement the UV / H2O2 process. The type of UV lamp is another important variable in this process, the selection of the waves generated by the lamp is a design parameter that defines the efficiency of the system. The medium pressure ultraviolet lamp (MP-UV) and the low pressure ultraviolet lamp (LP-UV) are the two types of lamps used in the UV/H2O2 system. The MP-UV lamp is usually the most widely used as it is capable of emitting a broad spectrum of waves much faster than the LP-UV lamp, allowing a rapid dissociation of peroxide radicals resulting in a direct photolysis that allows a faster degradation of the pollutants present in tannery effluents [110]. Temperature is an important factor in the UV / H2O2 system, at room temperature the reaction of the peroxide with the pollutants present in the tannery effluents is lower, hence it is required to accelerate the process by increasing the temperature (40 â—¦C and 60 â—¦C ) allowing the generation of OH from H2O2 and increasing the reactivity of these radicals towards pollutants [119].

Cavitation is a phenomenon that results in the generation of highly reactive free radicals, releasing large amounts of energy and creating intense turbulence in the liquid. highly reactive free radicals, releasing a large amount of energy and creating intense turbulence in the liquid, depending on the cavitation mode 4 different forms are distinguished: acoustic cavitation, hydrodynamic cavitation, cavitation, optical cavitation and particle cavitation. It has been reported that acoustic cavitation and hydrodynamic cavitation have proven to be effective techniques for the treatment of effluents. for the treatment of tannery effluents [110, 120]. The process parameters that affect acoustic cavitation are: ultrasonic power since the amount of OH generation and the energy dissipated depend on the ultrasonic power and the immersion depth of the probe, it is essential that the immersion depth of the probe is adequate for the propagation and dissipation of the wave. As for hydrodynamic cavitation, factors such as pH, temperature and inlet pressure affect the removal efficiency of parameters such as COD and color [121]

  1. Page 8, line 248 “advanced oxidation processes to electrochemical” this term is not clearly defined.. what does this mean? Please clarify it.

Answer: this sentence about electrochemical process was modified.

  1. Page 11, lines 262-263 “different microorganisms have been proposed as an efficient and economical alternative”. Please provide data to sustain this affirmation. I do not see any values which confirm that it is an economical alternative.

Answer: The sentence was modified.

These difficulties have been overcome in recent years, where different microorganisms have been proposed as an efficient alternative and an opportunity to establish bioremediation centers and economic biorefineries to recover resources and promote sustainable economic development for the treatment of wastewater and the removal of heavy metals from water [132,133] Studies on these processes use biological agents such as bacteria, microalgae and some fungi, among others [7,134].

  1. Page 13, line 319 “...transported to the cathode chamber where oxygen..”. This sentence is not complete.

Answer: the sentence was modified.

  1. Please delete mW/m2 from the head of table 3 and % from the values in column 2.

Answer: it was made.

  1. Page 15, lines 410-411..what about the accumulated heavy metals in the formed biomass. What is their environmental impact considering that they are concentrated in the biomass?

Answer: The data was included.

  1. Please insert a list of abbreviations. Also, the unit for the involved parameters needs to be added in accordance with the SI.

Answer: it was made.

Round 2

Reviewer 1 Report

The manuscript was enhanced following the suggestions. I consider that it can be accepted.

Reviewer 2 Report

Dear Authors, I appreciate your efforts in corrections to raise the quality level of your article. The manuscript has been revised following the comments and suggestions raised by reviewers. I think the content of the revised version of manuscript is meaningful and the paper is well organized.